# Maternal COVID-19 Vaccine May Reduce the Risk of MIS-C in Infants: A Narrative Review

**DOI:** 10.3390/vaccines10091454

**Published:** 2022-09-02

**Authors:** Chetna Mangat, Siva Naga Srinivas Yarrarapu, Gagandeep Singh, Pankaj Bansal

**Affiliations:** 1Department of Pediatrics, Mayo Clinic Health System, 733 W Clairemont Ave, Eau Claire, WI 54701, USA; 2Department of Internal Medicine, Monmouth Medical Center/RWJBH, Long Branch, NJ 07740, USA; 3Department of Family Medicine, Mayo Clinic Health System, 733 W Clairemont Ave, Eau Claire, WI 54701, USA; 4Department of Rheumatology, Mayo Clinic Health System, 1400 Bellinger Street, Eau Claire, WI 54701, USA

**Keywords:** COVID-19 vaccine, pregnancy, infant, MIS-C

## Abstract

COVID-19 infection in the pediatric population usually leads to a mild illness; however, a rare but serious complication of MIS-C has been seen in children. MIS-C usually presents 2–4 weeks after COVID-19 infection or exposure, and rare reports have been documented in neonates. Vaccinations for COVID-19 have been approved for children aged 6 months and above in the United States, and recent reports suggest significantly low prevalence and risk of complications of Multi-organ Inflammatory Syndrome (MIS-C) in vaccinated children compared to unvaccinated children. Vaccinations for COVID-19 are safe and recommended during pregnancy and prevent severe maternal morbidity and adverse birth outcomes. Evidence from other vaccine-preventable diseases suggests that through passive transplacental antibody transfer, maternal vaccinations are protective against infections in infants during the first 6 months of life. Various studies have demonstrated that maternal COVID-19 vaccination is associated with the presence of anti-spike protein antibodies in infants, persisting even at 6 months of age. Further, completion of a 2-dose primary mRNA COVID-19 vaccination series during pregnancy is associated with reduced risk for COVID-19–associated hospitalization among infants aged 6 months or less. Therefore, it can be hypothesized that maternal COVID-19 vaccination can reduce the risk of and severity of MIS-C in infants. In this article, we review the literature to support this hypothesis.

## 1. Introduction

Severe acute respiratory syndrome coronavirus-2 (SARS-CoV-2) causes relatively mild illness in children as compared to adults, and children usually recover without any adverse clinical course [1,2]. During this pandemic, an increasing incidence of Kawasaki-like disease has been seen in otherwise healthy children which is now termed multiorgan inflammatory syndrome (MIS-C). MIS-C is a rare but serious consequence of SARS-CoV-2 infection which occurs 2–5 weeks after SARS-CoV-2 infection or exposure [3]. Most of the cases are reported in children aged 8 to 12 years but young children and even neonates are not spared this severe multi-organ inflammatory disease [4]. Cardiac complications are reported in 50% of the cases and due to the complicated course, most children who develop MIS-C require care in the intensive care unit (ICU) [5,6]. The Centers for Disease Control and Prevention (CDC) started tracking MIS-C cases in May 2020 and since then, 8210 cases of MIS-C and 66 deaths due to MIS-C have been identified [7].

An increasing number of MIS-C cases in infants 0–6 months of age are being reported and most of the data is emerging in the form of case reports and case series. As per the latest systematic analysis, a total of 90 cases of MIS-C and MIS-N were identified in 0–6-month-old infants [8]. However, there is no formal definition of MIS-N, and this term is used by some clinicians for case identification in neonates born to SARS-CoV-2 positive mothers [8,9]. Both vertical and horizontal transmission of the SARS-CoV-2 virus is the proposed etiology for MIS-C in infants [8].

The COVID-19 vaccine is approved for children 6 months of age and above in the United States and it decreases the severe complications of SARS-CoV-2 infection, hospitalization, and death [10]. There is growing evidence that MIS-C incidence is lower in vaccinated adolescents [11,12,13]. Further, hospitalization rates are lower in vaccinated children, and infants 0–6 months of age born to vaccinated mothers [14]. COVID-19 vaccines have been approved for pregnant females, and have been proven to be safe and effective in preventing COVID-19 incidence and its complications in pregnant mothers [15]. Transfer of IgG antibodies to infants after maternal vaccination has been well-documented for other vaccines, and these antibodies can persist in the infant beyond the neonatal period, providing protective effects in the infant [16]. It can therefore be hypothesized that maternal vaccination for COVID-19 can lead to transplacental transfer of IgG antibodies to the infant, and render protection against not only COVID-19 but also its complications, such as MIS-C in infants. In this article, we aimed to review the available literature that evaluated: (1) transplacental transfer of antibodies to the newborn and during early infancy after maternal COVID-19 vaccination, and (2) the risk of COVID-related hospitalization and MIS-C incidence in vaccinated children and infants born to mothers vaccinated during pregnancy.

## 2. Materials and Methods

Our search was aimed at identifying studies (1) evaluating transplacental transfer of antibodies to the newborn following maternal COVID-19 vaccination, and (2) evaluating the impact of COVID-19 infection-related hospitalization and MIS-C incidence in vaccinated children and infants born to mothers who were vaccinated during pregnancy. Data for this review were identified by searches of MEDLINE, EMBASE, SCOPUS, Google Scholar, Science Citation Index, and references from relevant articles using the search terms “maternal”, “vaccine”, “vaccination”, “BNT162b2”, “Pfizer-BioNTech”, “mRNA-1273”, “mRNA”, “severe acute respiratory syndrome coronavirus 2”, “SARS-CoV-2”, “COVID-19”, “2019-nCoV” and “coronavirus”. Only articles published in English from inception to 18 May 2022 restricted to humans, and directly related to this review were included.

## 3. Results

### 3.1. Studies Evaluating Transplacental Transfer of Antibodies to Newborns and Young Infants after Maternal COVID-19 Vaccination

Our search yielded 19 studies that have thus far evaluated the transplacental transfer of antibodies after maternal vaccination (Table 1). Most of these case reports/case series and prospective cohort studies are reported from the United States and Israel. After a detailed review, all the studies demonstrated that anti-spike protein (anti-S) IgG antibodies were detected in the umbilical cord samples of neonates at birth after maternal vaccination. Trostel et al (2021) studied that these antibodies were positive in all the neonates at birth [17]. Interestingly, the fetal titers could reach the maternal level just after 15 days of the first dose of the COVID-19 vaccine, but neonates born to vaccinated parturients had higher antibody titers and prolonged protection compared to those born to infected parturients [18,19,20,21]. Vaccination in the third trimester and a booster dose in this trimester were also found to be associated with high maternal anti-spike IgG levels [22]. Six of the studies tested the neutralizing properties of transplacentally transferred maternal antibodies and revealed that the neutralizing capacity of neonatal sera was comparable to maternal sera, which could protect infants from SARS-CoV-2 infection and its complications [21,23,24,25,26].

The next question relates to the durability of these antibodies. Two of the studies evaluated the duration for which the transplacentally transferred maternal anti-S IgG antibodies were present in the infant. Shook et al. (2022) studied 77 vaccinated pregnant women and 12 pregnant women with SARS-CoV-2 infection during pregnancy and tested the durability of these antibodies in both of these groups [27]. Vaccinated parturients had high titers at birth, with respective high levels in cord blood samples. Furthermore, these antibodies were positive in 98% of infants at 2 months of age and declined to 57% at 6 months of age, but compared to infants born to parturients with SARS infection in pregnancy, only 8% of these infants had positive titers at 6 months of age. Mangat et al. (2021) also reported case reports in which antibodies were persistently positive at 6 months of age, but the titers were declining [28].

**Table 1 vaccines-10-01454-t001:** Summary of studies showing the transplacental transfer of antibodies to newborns and early infants after maternal COVID-19 vaccination.

Author & Year	Study Type	Location	Type of Vaccine	Study Groups	Time of Vaccine in Pregnancy	Measure	Results
Beharier, 2021 [18]	Prospective cohort	Israel	BNT162b2 (2 doses)	(a) Vaccinated parturients (*n* = 86); (b) PCR-confirmed SARS-CoV-2-infected during pregnancy (*n* = 65) (c) Unvaccinated non-infected parturients (*n* = 62)	34.5 ± 7.5 weeks	Anti-S and RBD antibodies in maternal blood and in umbilical cord blood (Sera Ig G and IgM)	(1) Vaccination elicited strong maternal humoral IgG response (2) Maternal titers in the fetus attained within 15 days following the first dose.
Collier, 2021 [25]	Prospective cohort	Israel	BNT162b2 or MRNA-1273 (2 doses)	(a) 30 pregnant, 16 lactating, 57 neither pregnant nor lactating vaccinated women (b) 22 pregnant and 6 non-pregnant unvaccinated women with SARS-CoV-2 infection	1st trimester (17%), 2nd trimester (50%), 3rd trimester (33%)	Immunogenicity of the mRNA vaccines in pregnant and lactating women: median RBD and anti-S IgG and IgA antibody titers	(1) Vaccination was immunogenic in pregnant women. (2) Vaccine-elicited antibodies were transported to infant cord blood and breast milk.
Kashani-Ligumsk, 2021 [19]	Prospective cohort	Israel	BNT162b2 (2 doses)	(a) Women who were vaccinated during pregnancy (*n* = 29) (b) Infected with SARS-CoV-2 during pregnancy (*n* = 29) (c) Women not infected and not vaccinated (*n* = 21)	3rd trimester	Titers of anti-S IgG antibodies to SARS-CoV-2 in umbilical cord blood in vaccinated pregnant women	Neonates born to vaccinated parturients had higher antibody titers and prolonged protection compared to those born to infected parturients.
Shook, 2022 [27]	Prospective cohort	United States	BNT162b2 or MRNA-1273	(a) 77 vaccinated pregnant mothers (b) 12 pregnant women with natural SARS-CoV-2 infection	27 weeks	Persistence of anti-S IgG in infants after vaccine vs. natural infection.	Most infants had anti-S IgG persistently positive at 6 months of life.
Nir, 2021 [20]	Prospective cohort	Israel	BNT162b2 (2 doses)	(a) 64 vaccinated parturient women (b) 11 parturient women who had COVID-19 during pregnancy	33.5 ± 3.2 weeks at second dosage	RBD IgG antibodies in maternal blood and in umbilical cord blood	Antibodies detected in cord blood, newborn dried blood spot, and breast milk samples were higher in vaccinated women than in COVID-19-recovered women.
Gray, 2021 [21]	Prospective cohort	United States	BNT162b2 or MRNA-1273 (2 doses)	(a) Vaccinated (84 pregnant, 31 lactating, and 16 non-pregnant women) (b) Unvaccinated pregnant women (*n* = 37) with natural SARS-CoV-2 infection	23.2 weeks(IQR 16.3–32.1)	Titers of severe acute respiratory syndrome coronavirus 2 spike and receptor-binding domain IgG, IgA, and IgM were quantified in participant sera and breastmilk	(1) Robust IgG titers were observed across pregnant, lactating, and nonpregnant women, and were significantly higher than in pregnant women with previous SARS-CoV-2 infection. (2) Immune transfer to neonates occurred via placenta and breastmilk.
Trostle, 2021 [17]	Prospective case series	United States	BNT162b2 or MRNA-1273 (at least 1 dose)	36 vaccinated pregnant women	1st trimester (6%), 2nd trimester (83%), 3rd trimester (11%)	Positive Anti-S IgG antibodies and negative anti-N IgG antibodies in maternal blood and umbilical cord blood	(1) Transplacental antibody transfer following vaccination during pregnancy. (2) 100% of cord blood specimens had high levels of anti-S antibodies.
Rottenstreich, 2021 [29]	Prospective case series	Israel	BNT162b2 (at least 1 dose)	20 vaccinated pregnant women	3rd trimester	Anti-S and RBD IgG antibodies in maternal blood and umbilical cord blood	(1) Vaccination during pregnancy was immunogenic, with transplacental antibody transfer (2) A positive correlation between maternal and cord blood antibody concentration was seen.
Mithal, 2021 [30]	Prospective case series	United States	BNT162b2 or mRNA-1273	27 vaccinated pregnant women	33 ± 2 weeks	Transfer of SARS-CoV-2 IgG antibodies to infants	(1) Pregnant women vaccinated during the 3rd trimester had transplacental antibody transfer. (2) Infant IgG levels were about equal to maternal levels.
Prabhu, 2021 [31]	Prospective case series	United States	BNT162b2 or MRNA-1273 (at least 1 dose)	122 vaccinated pregnant women	NA	Anti-S IgG antibodies in maternal blood and umbilical cord blood	(1) All women and cord blood samples, except for one, had detectable IgG antibodies by 4 weeks after vaccine dose 1. (2) Timing between vaccination and birth may be an important factor to consider in vaccination strategies of pregnant women.
Yang, 2022 [22]	Retrospective cohort	United States	BNT162b2 mRNA-1273 or Johnson	1359 vaccinated pregnant women	Both in pre-pregnant state and pregnancy	Transfer of maternal anti-S IgG in cord blood.Assess the association of prior infection and vaccine booster on anti-S IgG levels	(1) Maternal anti-S IgG antibodies were detectable at delivery regardless of timing of vaccination (2) Highest antibody levels detected with 3rd trimester vaccination. (3) A booster dose in the 3rd trimester was associated with maternal anti-spike IgG levels greater than 3rd trimester vaccination in women with/without a history of SARS-CoV-2 infection.
Cassaniti, 2021 [24]	Retrospective cohort	Italy	BNT162b2 (2 doses)	(a) 2 vaccinated pregnant women (b) 7 non-vaccinated controls with SARS-CoV-2 infection during pregnancy	31^+4^ and 27^+6^ weeks	anti-SARS-CoV-2 spike IgG and IgA antibodies in pregnant women and newborns	(1) Transplacental transfer of anti-SARS-CoV-2 IgG at delivery after infection or vaccination. (2) Median neutralizing antibody titer was twofold reduced in newborns with respect to mothers.
Gloeckner, 2021 [26]	Retrospective cohort	Germany	BNT162b2 or MRNA-1273 after a prime vaccination with Oxford–AstraZeneca ChAdOx1	(a) 3 vaccinated pregnant women (b) 25 vaccinated non-pregnant controls	NA	anti-spike IgG antibody kinetics in pregnant women in comparison to their newborns, as well as to a healthy nonpregnant control group	(1) Vaccination-induced immunogenic response in pregnant women showed no significant difference compared with non-pregnant controls 16 weeks after the initial vaccination. (2) Vaccination induced a strong passive humoral immunity in the newborns.
Zdanowski, 2021 [32]	Retrospective case series	Poland	BNT162b2 (2 doses)	16 vaccinated pregnant women	31.8 ± 2.1 weeks	Anti-S IgG antibodies in maternal blood and umbilical cord blood	(1) Maternal immunization provided neonatal protection through the transplacental transfer of antibodies. (2) Antibody transfer was correlated with the time from vaccination to delivery.
Douxfils, 2021 [23]	Case report	Belgium	BNT162b2 (2 doses)	1 vaccinated pregnant woman	25 weeks	anti-S IgG antibodies in umbilical cord blood compared to maternal blood	(1) Successful maternal–fetal transfer of neutralizing antibodies. (2) Levels of neutralizing antibodies were 5-fold higher in the umbilical cord than in the maternal blood.
Gill, 2021 [33]	Case report	United States	BNT162b2 (2 doses)	1 vaccinated pregnant woman	32^+6^ weeks	IgG SARS-CoV-2 antibodies in maternal blood and umbilical cord blood	Vaccination in pregnancy produced a robust immune response for the patient, with subsequent transplacental transfer of neutralizing antibodies.
Mangat, 2021 [28]	Case report	United States	BNT162b2	1 vaccinated pregnant woman	22 and 26 weeks	Serial anti-S IgG antibody titers in infant	Antibodies in preterm infants were persistently positive at 6 months of age.
Mehaffey, 2021 [34]	Case report	United States	BNT162b2 (2 doses)	1 vaccinated pregnant woman	29 and 32 weeks	IgG SARS-CoV-2 antibodies in maternal and umbilical cord blood	Vertical transmission of IgG SARS-CoV-2-specific antibodies from a vaccinated mother to her son with no evidence of prior infection.
Paul, 2021 [35]	Case report	United States	MRNA-1273 (1 dose)	1 vaccinated pregnant woman	36^+3^ weeks	IgG SARS-CoV-2 in umbilical cord blood after maternal vaccination	SARS-CoV-2 IgG antibodies were detectable in a newborn’s cord blood sample after only a single dose of the vaccine.

### 3.2. Studies Evaluating the Risk of COVID-19-Related Hospitalization and MIS-C Incidence in Vaccinated Children and Infants Born to Mothers Vaccinated during Pregnancy

Our search yielded 10 studies in this section, including 5 case-control studies, 3 population-based surveillance studies, 1 prospective cohort, and 1 patient registry. Most of the studies are case-control studies from the United States.

Between June 1 and September 30, 2021, during Delta variant predominance, 2 doses of BNT162B2 were effective in reducing 93% of COVID-19-related hospitalizations among patients aged 12–18 years [36]. Comparison of the data during Delta and Omicron predominance showed that COVID-19 vaccination decreased the hospitalization rate during the Omicron surge from 93 to 79% in the 12–18 age group, whereas vaccine effectiveness (VE) against hospitalization among children 5 to 11 years of age was 68%, for which the data is available only during the Omicron surge [37]. From April 2021 to January 2022, VE against hospitalization in the 5-to-11-year age group was 78% but VE was reduced in older children when the 2nd dose of vaccine was given >150 days from infection [38]. Furthermore, completion of 2 doses of mRNA COVID-19 vaccine during pregnancy showed VE of 61% against hospitalization in infants 0–6 months of age [14]. On the other hand, an increased hospitalization rate was noted in children 1–4 years of age during the Omicron surge, as the vaccination was not available for this age group during that time [39].

Receipt of 2 doses of BNT162b2 vaccine showed protection against MIS-C, which is a life-threatening complication of SARS-CoV-2 infection in children aged 12–18 years (July–December 2021) with estimated effectiveness of 91% against MIS-C, and along with this, MIS-C complications were also decreased in vaccinated children, and all MIS-C patients requiring life support were unvaccinated [11]. Studies from France reported similar findings, in which decreased MIS-C likelihood was noted after the first dose of the COVID-19 vaccine (From 1 September 2021 to 31 October 2021) [12]. Oduldali et al. (2022) reported that the risk of MIS-C in children 12–17 years of age in France was significantly lower after vaccination (1.5 per 1,000,000 doses of COVID-19 vaccine) as compared to the risk of MIS-C due to infection by SARS-CoV-2 (113 per 1,000,000 infected by SARS-CoV-2) [13].

Approved vaccine for children 12–18 is available in the USA since May 2021, and for children 5–11 years of age it has been available since November 2021 [40]. Vaccination has been found to be safe and effective and is being administered under the most intensive vaccine safety surveillance effort in history. Based on benefit-risk assessment, it’s estimated that approximately 1000 COVID-19 hospitalizations may have been prevented during the Omicron surge among fully vaccinated children ages 5–11 years, in the United States [41] (Table 2).

At the time of writing this paper, no study is available that has examined MIS-C in vaccinated younger children aged 5–11 years, although the vaccine has been available for this group since November 2021 in the United States. Risk of MIS-C has also not been examined yet in infants born to vaccinated mothers.

## 4. Discussion

CDC defines MIS-C as an individual aged less than 21 years presenting with fever, laboratory evidence of inflammation, and evidence of clinically severe illness requiring hospitalization, with more than two organ involvement with no alternative plausible diagnoses, and with current or recent SARS-CoV-2 infection or COVID-19 exposure within the 4 weeks prior to the onset of symptoms. [44] Most cases of MIS-C are clustered in the 8–11-year age group and the number of cases in neonates and infants reported thus far has been low [7] In a systematic meta-analysis, DeRose et al. (2022) analyzed 48 studies, which include 29 case reports, 6 case series and 13 cohort studies [8]. They identified 25 papers, which included a total of 32 cases of MIS-C in the form of case studies in infants aged 0–6 months. Additionally, 31 infants 0–6 months of age with MIS-C were identified in 12 existing cohort studies, which included a total of 1062 children 0–18 years old. The youngest case of MIS-C reported was in a 7-day-old infant [45]. MIS-N is the definition proposed by some clinicians for infants born to mothers with SARS-CoV-2 infection, but there is no formal definition of MIS-N. Based on this proposed definition, Del Rose et al. (2022) identified 33 neonates with MIS-N and interestingly, only 18.2% had fever as compared to fever as a major criterion in the diagnosis of MIS-C in children [8]. The true incidence of MIS-C is not known. Based on a multicenter study of a large cohort in the United States, the cumulative incidence of MIS-C in those younger than 21 years of age was 2.1 per 100,000 persons, with a mortality rate of 1.4% [4]. Incidence in newborns and infants is unknown due to scarcity of data and difficulty in diagnosis as newborns could be afebrile, and due to many overlapping symptoms, it can be difficult to differentiate MIS-C from sepsis in early infancy [8]. Although there are only a few cases reported so far, risk of cardiac complications and critical illness is high in this population, with very a high mortality of 12.5% reported in infants 0–6 months of age [4,8].

MIS-C is temporally associated with current or prior SARS-CoV-2 infection, but the exact pathogenesis of MIS-C is not fully understood. Available literature proposes the multilineage activation of the immune system, which includes both innate and adaptive signatures [46]. Viral load at the time of MIS-C diagnosis is lower, and it has been hypothesized to be an autoimmune and hyperinflammatory response to the initial infection. High levels of inflammatory cytokines, including interferon alpha and gamma, and interleukin (IL) -1β, IL-6, IL-8, IL-10, and IL-17, are reported but no pathognomonic marker has yet been identified [47,48]. Decreased expression of angiotensin convertase enzyme (ACE) receptors in early infancy and predominantly stronger innate immune response with less adaptive immune development are the proposed reasons behind the decreased incidence of severe COVID-19 infection and eventually, the development of MIS-C in infants.

The safety and efficacy of COVID-19 vaccination in children > 5 years of age has been well documented. Further, a multistate study from the US hospital network by Zambrano et al. (2022) revealed that receipt of 2 doses of the Pfizer-BioNTech vaccine was associated with protection against MIS-C in patients aged 12–18 years. The likelihood of developing MIS-C was 91% less in vaccinated children, and vaccinated children who developed MIS-C were less likely to develop respiratory or cardiac complications [11]. Similar studies from France also revealed a lower incidence of MISC-C in vaccinated children [12,13].

Our review identified several studies that have documented transplacental transfer of antibodies following maternal COVID-19 vaccination and a reduced risk of hospitalization related to COVID-19 in infants 0–6 months of age. Further, these antibodies can persist in infants up to 6 months of age, after which, a decline in their titers was reported [14,28]. It is unknown if the risk of COVID-19 and its complications increases corresponding to the decline in titers of these antibodies after 6 months or not. The risk of MIS-C in infants born to vaccinated mothers has not been extensively evaluated. Maternal vaccination leads to anti-spike protein IgG antibody production in maternal circulation, which then gets passively transferred to the fetus through transplacental transport and is detected in newborns after birth and early infancy and has a neutralizing effect against COVID-19 infection and its complications, including MIS-C [17,23,28,31] (Figure 1).

Vaccination for children in the 6 months through 4 years age group was recently approved, but so far, it is unknown if and when a vaccine for infants 0–6 months will be available. Our review supports the evidence that maternal COVID-19 vaccination could provide protection to the infant in the first 6 months of life, as seen with other recommended vaccines during pregnancy, e.g., Tdap and influenza, which have been successful in reducing the morbidity and mortality associated with these infections in infants [16].

Our review has many limitations. Due to heterogeneity of the available studies, a quantitative analysis could not be performed. Since the search was limited to English only and was not systematic, some studies could have been missed. Most of the available studies were case reports or case series, and thus, the level of evidence is low. Further, well-conducted randomized clinical trials can strengthen the evidence and provide clearer information about the risk of MIS-C in infants following maternal vaccination for COVID-19.

## 5. Conclusions

Maternal vaccination for COVID-19 may be effective in rendering passive immunity to infants under 6 months of age. There is emerging evidence that it can prevent SARS-CoV-2 infection and its complications, and hence, the life-threatening consequence of MIS-C in infants < 6 months. The data is still premature and further studies can elicit the exact impact and timing of vaccination during pregnancy on the risk of MIS-C in infants.

## Figures and Tables

**Figure 1 vaccines-10-01454-f001:**
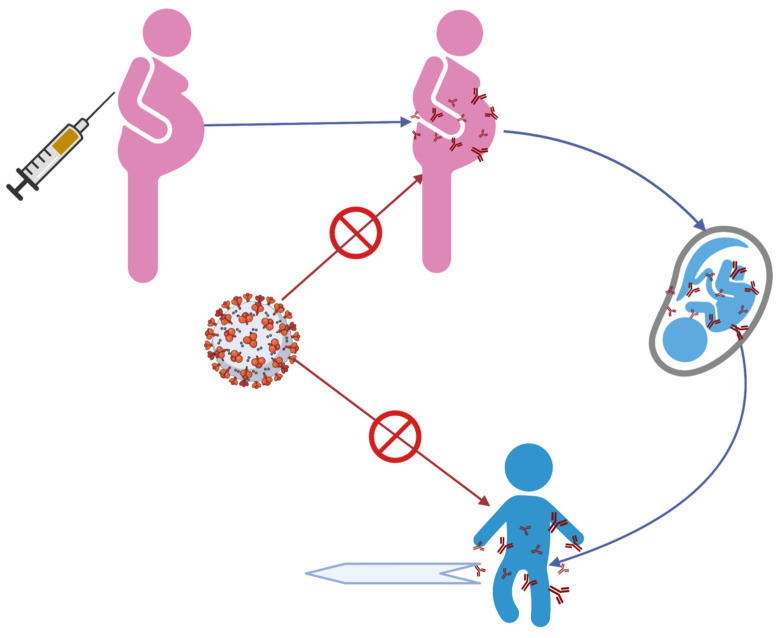
Proposed hypothesis for reducing the risk of MIS-C in infants after material COVID-19 vaccination. (Created with BioRender.com) access date 25th May 2022.

**Table 2 vaccines-10-01454-t002:** Summary of studies showing the decrease in COVID-19-related hospitalization and MIS-C incidence in vaccinated children and infants born to mothers vaccinated during pregnancy.

Author	Country	Type of Study	n: Study Group vs. Control Group (if Any)	Age	Intervention	Study Period	Measure	Results:
Olson, 2022 [36]	United States	Case-control	179 cases (SARS-CoV-2 test positive) vs. 285 controls (SARS-CoV-2 test negative)	12–18 years	2 Doses of BNT162b2	June– September 2021	Vaccine effectiveness (VE) against COVID-19-associated hospitalization	VE = 93% (95% CI, 83–97%); predominant variant: Delta
Klein, 2022 [38]	United States	Case-control	9640 cases (SARS-CoV-2 test positive) vs. 31,276 controls (SARS-CoV-2 test negative)	5–17 years	2 Doses of BNT162b2	April 2021–January 2022	Vaccine effectiveness against COVID-19-associated hospitalizations (ED/UC encounters)	Age 5–11 years; VE = 46%. Age 12–17 years; VE ≥ 90% (protection within 149 days of second dose administration.)
Price, 2022 [37]	United States	Case-control	1185 cases (SARS-CoV-2 test positive) vs. 1627 controls (SARS-CoV-2 test negative)	5–18 years	2 Doses of BNT162b2 m RNA vaccine	July–February 2022	Vaccine effectiveness against COVID-19-associated hospitalization, critical COVID-19 and non-critical COVID-19	Age 12–18 years; VE = 93% (95% CI, 89–95%) at 2–22 weeks following vaccination and VE = 92% (95% CI, 80–97%) at 23 44 weeks, predominant variant: Delta Age 12–18 years; VE = 40% (95% CI, 9–60%) against COVID-19-associated hospitalization, 79% (95% CI, 51–91%) against critical COVID-19, and 20% (95% CI, −25–49%) against noncritical COVID-19 (median interval of 162 days following vaccination) predominant variant: Omicron Age 5–11 years, VE = 68% (95% CI, 42–82%; median interval of 34 days since vaccination). predominant variant: Omicron
Halasa, 2022 [14]	United States	Case-control	176 cases (SARS-CoV-2 test positive) vs. 203 controls (SARS-CoV-2 test negative)	0–6 months	Maternal 2-dose mRNA vaccination	July 2021–January 2022	Vaccine effectiveness against COVID-19-associated hospitalization	Infants aged < 6 months; VE = 61% (95% CI, 31–78%). VE = 32% (95% CI, −43–68%), (vaccination in first 20 weeks), and VE = 80% (95% CI, 55–91%), (vaccination later in pregnancy: 21 weeks through 14 days before delivery)
Zambrano, 2022 [11]	United States	Case-control	102 cases (diagnosed with MIS-C) vs. 181 controls (no evidence of SARS-CoV-2 infection)	12–18 years	2 Doses of Pfizer-BioNTech vaccine in 12–18 year-olds	July 2021–January 2022	Vaccine effectiveness against MIS-C	VE = 91% (95% CI = 78–97%)
Fowlkes, 2022 [42]	United States	Prospective cohort	1364 participants	5–15 years	2 Doses of BNT162b2	July 2021–February 2022	Vaccine effectiveness against COVID-19-associated hospitalization	Age 5–11 years, VE = 31% (95% CI, 9–48%) (14–82 days after dose 2 administration) against Omicron infection Age 12–15 years, VE = 87% (95% CI, 49–97%) (14–149 days after dose 2 administration), variant: Delta infection, and 59% (95% CI, 22–79%), variant: Omicron infection
Shi, 2022 [43]	United States	Population-based surveillance	1475 (laboratory- confirmed COVID-19)	5–11 years	1–2 Doses of BNT162b2	March 2020–February 2022	COVID-19-associated hospitalizations in unvaccinated children	Cumulative hospitalization rates among unvaccinated children were 2.1 times that of (19.1) vaccinated.
Ouldali, 2022 [13]	France	Population-based surveillance	NA	12–17 years	2 Doses of mRNA vaccine	15 June 2021 and 1 January 2022	Vaccine effectiveness against MIS-C	MIS-C incidence = 2.9 per 1,000,000 in vaccinated children MIS-C incidence = 113 per 1,000,000 in children infected with SARS-CoV-2
Marks, 2022 [39]	United States	Population-based Surveillance	2818 (laboratory- confirmed COVID-19)	12–17 years	1–2 Doses of BNT162b2	July 2021–January 2022	Vaccine effectiveness against COVID-19 associated hospitalization	Monthly hospitalization rates in unvaccinated adolescents aged 12–17 years (23.5) was 6 times as much as in fully vaccinated adolescents (3.8). Increased rate of hospitalization in 0–4-year-olds among unvaccinated
Levy, 2021 [12]	France	Patient Registry	107 (diagnosed with MIS-C)	12–18 years	1–2 Doses of mRNA vaccine	September 2021 and October 2021	Vaccine effectiveness against MIS-C	HR for MIS-C = 0.09 (95% CI, 0.04–0.21%; *p* < 0.001)

## Data Availability

Not applicable.

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
