# Peer review of "Maternal COVID-19 Vaccine May Reduce the Risk of MIS-C in Infants: A Narrative Review"

_vaccines, 2022, doi:10.3390/vaccines10091454_

Round 1

Reviewer 1 Report

This is a good summary review of the available literature is this area of research, which is limited. Overall it is very well written and presented and it does add very good value to the understanding of this vital issue in a timely manner.

Manuscript ID: Vaccines-1893827

Type of manuscript: Review

Title: Maternal COVID-19 Vaccine May Reduce the Risk of MIS-C in Infants: A Narrative Review.

Authors: Chetna Mangat, Siva Naga Srinivas Yarrarapu, Gagandeep Singh, Pankaj Bansal * Submitted to section: COVID-19 Vaccines and Vaccination.

Comments/suggestions:

Abstract

Very good and to the point.

Minor suggestion – full form of MIS-C would be useful here as well.

Introduction

Overall good - with limited literature available for such a review manuscript. 

Line 50 “a total of 90” would be better.

Lines 53- 54 – is there a reference for this statement ?.

Lines 58 and 64 – typing error - full stop missing or placed after the reference and this are common errors through the manuscript which need to be corrected.

CDC definition of MIS-C would be more appropriate here than starting the discussion. In discussion it can be started as “Based on the CDC definition……” 

Materials and methods:

Appropriate.

Results

Line 94 – Trostel et al. – adding the year of publication in ( ) is the norm for most journals. This again needs to be fixed on other areas of this manuscript.

Line 99 – starting the sentence with “Six” instead of 6 would be better.

Line 109 – “which further declined … I think this “further” is not appropriate here.

Lines 136 – 140 – Not clear, needs rewording or breaking the sentence to make it clear.

Line 153 – not clear

Line 155 – 1,000 – out of what population/hospital/area/region/state/country?

Tables:

Very good summaries presented on both tables but they would need be presented/formatted better.

Table titles could be – Summary of studies …. 

Discussion

Good discussion and does make the limitations of this review clear.

Author Response

This is a good summary review of the available literature is this area of research, which is limited. Overall it is very well written and presented and it does add very good value to the understanding of this vital issue in a timely manner.

Manuscript ID: Vaccines-1893827

Type of manuscript: Review

Title: Maternal COVID-19 Vaccine May Reduce the Risk of MIS-C in Infants: A Narrative Review.

Authors: Chetna Mangat, Siva Naga Srinivas Yarrarapu, Gagandeep Singh, Pankaj Bansal * Submitted to section: COVID-19 Vaccines and Vaccination.

Comments/suggestions:

Abstract

Very good and to the point.

Minor suggestion – full form of MIS-C would be useful here as well.

We have added full form of MIS-C in the abstract.

Introduction

Overall good - with limited literature available for such a review manuscript. 

Line 50 “a total of 90” would be better.

We have made this change. It appears there is a disconnect in the line numbers, This line on the manuscript is line 45 and not line 50. We have made the change here as suggested by the reviewer.

Lines 53- 54 – is there a reference for this statement ?.

Given the disconnect in the line numbers, the reviewer here is likely referring to the statement on the proposed mechanisms of MIS-C which is line 47-48.  We have provided a reference for this statement (Vivanti AJ, Vauloup-Fellous C, Prevot S, et al. Transplacental transmission of SARS-CoV-2 infection. Nat Commun. 2020;11(1):3572. Published 2020 Jul 14. doi:10.1038/s41467-020-17436-6).

Lines 58 and 64 – typing error - full stop missing or placed after the reference and this are common errors through the manuscript which need to be corrected.

We have corrected this error everywhere in the manuscript as suggested by the reviewer.

CDC definition of MIS-C would be more appropriate here than starting the discussion. In discussion it can be started as “Based on the CDC definition……” 

We have made this change as suggested by the reviewer, and moved the case definition of MIS-C from the discussion section to the introduction section.

Materials and methods:

Appropriate.

Results

Line 94 – Trostel et al. – adding the year of publication in ( ) is the norm for most journals. This again needs to be fixed on other areas of this manuscript.

We have made this change as suggested by the reviewer in all areas of the manuscript.

Line 99 – starting the sentence with “Six” instead of 6 would be better.

We have made this change as suggested by the reviewer.

Line 109 – “which further declined … I think this “further” is not appropriate here.

We have made this change as suggested by the reviewer.

Lines 136 – 140 – Not clear, needs rewording or breaking the sentence to make it clear.

We have changed this entire section to clarify the sentence.

Line 153 – not clear

We have made changes to clarify the sentence.

Line 155 – 1,000 – out of what population/hospital/area/region/state/country?

Based on this study (reference 41), this is the total number of hospitalizations that have been likely prevented by vaccination of children aged 5-11 in the US. We have added “in the United States” to this sentence.

Tables:

Very good summaries presented on both tables but they would need be presented/formatted better.

Table titles could be – Summary of studies …. 

We have changed the titles of the tables. We will be happy to change the format of the table as per the publisher guidelines/suggestions.

Discussion

Good discussion and does make the limitations of this review clear.

Reviewer 2 Report

This is an interesting review and I have only a few comments on it;

Authors need to define the study selection i.e. inclusion criteria. It seems that only studies including pregnant women who are going to deliver the baby, or those with neonates who are currently breastfeeding their babies were included in this review. 

This section may be divided into subheadings. This sentence can be written as "This section is divided into subheadings. I"

Please provide more information in the discussion section about the benefits of maternal transfer of antibodies for other infectious diseases. There is a need to urge the health authorities to consider maternal vaccination, specifically those delivering soon, or for those breastfeeding their babies. This will definitely assist the authorities to protect the children from future exposure.

Is there any specific reason that this review is written in a narrative way, rather than systematic one?

Author Response

This is an interesting review and I have only a few comments on it;

Authors need to define the study selection i.e. inclusion criteria. It seems that only studies including pregnant women who are going to deliver the baby, or those with neonates who are currently breastfeeding their babies were included in this review. 

The study selection has been mentioned as “identifying studies (1) evaluating transplacental transfer of antibodies to the newborn following maternal COVID-19 vaccination, and (2) evaluating the impact of COVID-19 infection-related hospitalization and MIS-C incidence in vaccinated children and infants born to mothers who were vaccinated during pregnancy”. Based on the second inclusion criteria, we also reviewed the studies that examined the risk of MIS-C in vaccinated children compared to unvaccinated children.

This section may be divided into subheadings. This sentence can be written as "This section is divided into subheadings. I"

We apologize for this error, this sentence was a “carry over” from the format for writing manuscript from the journal, and should have been removed in the manuscript. We have removed this sentence as this was an error.

Please provide more information in the discussion section about the benefits of maternal transfer of antibodies for other infectious diseases. There is a need to urge the health authorities to consider maternal vaccination, specifically those delivering soon, or for those breastfeeding their babies. This will definitely assist the authorities to protect the children from future exposure.

Our article focuses on the benefits of vaccination for COVID-19 during pregnancy, and reviewing studies examining the benefit of vaccination in prevention of MIS-C. We believe that a detailed or more information about other infectious diseases will be beyond the scope of this article, and can dilute the impact of our review article. We have briefly mentioned other diseases in the introduction section “Transfer of IgG antibodies to infants after maternal vaccination has been well documented for other vaccines, and these antibodies can persist in the infant beyond the neonatal period, providing protective effects in the infant. [16]”.

Is there any specific reason that this review is written in a narrative way, rather than systematic one?

Due to limitations in resources and time constraints, we were unable to perform a systematic review of the literature, which would have required extensive resources to meet PRISMA guidelines. We agree that a systematic review would have more impact than a narrative review and have highlighted this in the study limitation section. 
